# Excess winter mortality in Finland, 1971–2019: a register-based study on long-term trends and effect modification by sociodemographic characteristics and pre-existing health conditions

Ulla Suulamo  ,[1,2,3] Hanna Remes,[1,2] Lasse Tarkiainen,[1,2] Michael Murphy,[4] Pekka Martikainen[1,2,5]

For numbered affiliations see end of article.

**Correspondence to**
Ulla Suulamo;
ulla.suulamo@helsinki.fi

## ABSTRACT

**Objectives** Excess winter mortality is a well-established phenomenon across the developed world. However, whether individual-level factors increase vulnerability to the effects of winter remains inadequately examined. Our aim was to assess long-term trends in excess winter mortality in Finland and estimate the modifying effect of sociodemographic and health characteristics on the risk of winter death.

**Design** Nationwide register study.

**Setting** Finland.

**Participants** Population aged 60 years and over, resident in Finland, 1971–2019.

**Outcome measures** Age-adjusted winter and non-winter death rates, and winter-to-non-winter rate ratios and relative risks (multiplicative interaction effects between winter and modifying characteristics).

**Results** We found a decreasing trend in the relative winter excess mortality over five decades and a drop in the series around 2000. During 2000–2019, winter mortality rates for men and women were 11% and 14% higher than expected based on non-winter rates. The relative risk of winter death increased with age but did not vary by income. Compared with those living with at least one other person, individuals in institutions had a higher relative risk (1.07, 95% CI 1.05 to 1.08). Most pre-existing health conditions did not predict winter death, but persons with dementia emerged at greater relative risk (1.06, 95% CI 1.04 to 1.07).

**Conclusions** Although winter mortality seems to affect frail people more strongly—those of advanced age, living in institutions and with dementia—there is an increased risk even beyond the more vulnerable groups. Protection of high-risk groups should be complemented with population-level preventive measures.

## INTRODUCTION

In temperate climates, mortality rates tend to be higher in winter compared with other times of the year. Although studies on long-term trends indicate that this winter excess has weakened over time,[1–4] it remains a

## STRENGTHS AND LIMITATIONS OF THIS STUDY

⇒ We used high-quality, individual-level register data spanning nearly 50 years with minimal attrition and no recall bias.
⇒ The use of register data allowed us to examine risk factors at the individual level.
⇒ We used data on healthcare visits and reimbursement for medicine costs to examine whether those with pre-existing health conditions are particularly vulnerable.
⇒ Using only the underlying cause of death may underestimate the total impact of respiratory infections and influenza-like illnesses on excess winter mortality.

significant public health burden in many high-income countries, with winter increases in deaths of up to 28%.[5] This burden—likely driven both by the direct effects of winter conditions on the most frail individuals, as well as increased seasonal pressure on the healthcare system and knock-on effects on other patients—may grow even greater in ageing populations, considering that the risk of winter death increases with age and is highest among the oldest old.[6 7]

The increased risk of dying in winter is attributed to multiple factors, most importantly influenza-like communicable illnesses[8 9] and low temperatures.[10] Consistent with these proximate risk factors, a winter peak in mortality is apparent for most major causes of death, but the largest seasonal fluctuations are seen for circulatory and respiratory diseases.[11] While such external effects drive the overall seasonal pattern, several non-seasonal risk factors have been found to importantly modify an individual's vulnerability to the effects of winter. Previous evidence is clear in relation to age:

excess winter mortality especially affects older people, gradually increasing with age.[12] Besides age, factors such as low income, inadequate housing conditions and fuel poverty[13] have been associated with increased adverse health outcomes in winter, especially in countries with thermally inefficient housing stocks such as Portugal[14] and the UK.[15] Although based on limited evidence, living arrangements are another potentially important factor in differential mortality, and some previous findings suggest a higher risk of excess winter mortality for those living alone[12] or in institutions.[16–18]

The importance of non-seasonal risk factors is further demonstrated by cross-country comparisons, which suggest that absolute winter temperatures matter little, as countries with milder winters generally experience much greater wintertime excess than countries with colder climates.[5 19] These somewhat counterintuitive findings have led investigators to suggest that populations' ability to protect themselves is what ultimately determines the extent of winter mortality,[20] and indicate that winter excess deaths are, to an important degree, preventable.[5 7 21]

Despite the avoidable character of a substantial fraction of excess winter mortality, uncertainty still exists whether the winter excess mainly affects frail individuals who are at greater risk of dying soon in any case. Those with pre-existing health conditions are often considered particularly vulnerable, but empirical evidence to confirm this remains scarce. One previous study[22] assessing the determinants of vulnerability to excess winter mortality found evidence for an increased risk of winter death only for those with a history of respiratory illness but not for those with other health conditions. Addressing whether excess winter mortality mainly affects the frail remains thus an important area of further inquiry with potentially important policy implications: identifying those most affected may help evaluate how preventable these excess deaths are and guide the development of targeted prevention measures.

In international comparison, northern European countries with relatively cold winters show lower levels of excess winter mortality which has been interpreted to arise from a combination of socioeconomic, housing and behavioural factors.[19] Despite lower levels of excess, the winter period is nevertheless acknowledged to pose a range of health challenges,[23] and population ageing makes it likely that these challenges will become more pressing. Little is, however, known about how excess winter mortality differs by individual-level social and health characteristics in a societal context with high-quality housing at all income levels and universal access to healthcare services. Using individual-level data on the total population, we examined long-term trends in excess winter mortality in Finland at ages 60 and above in 1971–2019 and assessed differences in the winter excess by cause of death and the modifying effect of sociodemographic characteristics and pre-existing health conditions in 2000–2019.

## METHODS

### Data

The study was based on individual-level data for the total population of Finland from August 1971 to July 2019. These data were drawn from censuses and population registers that Statistics Finland had linked to cause-of-death records and healthcare registers using personal identification numbers. For 1971–1987, sociodemographic information was obtained from the end-of-year census records (1970/1975/1980/1985) and from the year 1988 onwards from the end-of-year population records. We restricted the analysis to individuals aged 60 years and over at the end of each year from 1971 to 2018.

We defined winter as December–March following previous studies in the northern hemisphere.[13] These months also meet the definition of thermal winter in Finland with daily mean temperatures below 0°C (32°F). On average, in central and southern Finland, where most of the population resides, thermal winter begins around late November and ends in late March. Non-winter months were defined as the 4 months on each side of this winter period (August–November, April–July). Following this definition, our study period to analyse trends over time ran from August 1971 to July 2019, 48 years in total. For each year, we followed mortality from the beginning of August to the end of July next year.

For the more recent 19-year period August 2000–July 2019, we assessed whether and how excess winter mortality was modified by sex, age, living arrangement, income, region and pre-existing health conditions. Age was categorised into groups 60–69, 70–79, 80–89 and 90 years and over. Living arrangement distinguished between those living in households with at least one other person, alone or in institutions. Household disposable income was categorised into quartiles in the population aged 60 and over and adjusted for the household composition.[24] For persons living in institutions, we used personal disposable income. Information on region of residence (n=19) was aggregated into four larger regions (coastal, southern, central and northern) according to average winter temperatures. Living arrangement, income and region were measured at the end of the preceding calendar year, whereas age corresponded to attained age in July of each year.

We used 11 dichotomous indicators of pre-existing health conditions: dementia, mental disorders, cancer, diseases of the nervous system, diabetes, diseases of the circulatory system, diseases of the respiratory system, alcohol-related causes, injuries due to accidents and violence, other health conditions and an overall measure of any health condition. These conditions were identified from two sources covering the years 1999–2017: the healthcare register of the Finnish Institute for Health and Welfare and the medication reimbursement register of the Social Insurance Institution of Finland. We included data on inpatient and specialised outpatient care episodes that occurred during the preceding calendar year (at least 7 months before the start of mortality follow-up)

and for which any of the aforementioned conditions were recorded as the primary discharge diagnoses. The health conditions were classified based on the International Classification of Diseases (ICD) 10th Revision. Information on the right to special reimbursement for medicine costs due to diagnosed chronic medical conditions during the preceding calendar year was collected from the medication reimbursement register of the Social Insurance Institution of Finland following their disease and medication classification. For exact coding of care episodes and reimbursement categories, see online supplemental table S1. A pre-existing health condition was considered present if there was a record in either of the two registers and each individual could contribute to several categories.

In addition to all-cause mortality, we assessed winter excess mortality separately for deaths due to diseases of the circulatory system (ICD-10 codes I00–I425, I427–I99), diseases of the respiratory system (J00–J64, J66–J99), dementia (F01, F03, G30, R54), other internal causes and external causes (V01–X44, X46–Y89). The coding of causes of death was based on Statistics Finland's cause of death classification, which is harmonised across different revisions of the ICD.

### Patient and public involvement
All data were collected for routine administrative purposes and, therefore, public involvement was not feasible.

### Statistical analysis
To obtain person-days at risk for each winter and non-winter period, we created an expanded dataset by splitting each year on 1 April and 1 December. Deaths were allocated to the appropriate winter and non-winter periods using exact dates of death.

To assess trends over time, we collapsed the expanded dataset by summing the death and person-time counts over sex, age, year, and winter and non-winter periods. We then calculated and plotted age-adjusted annual winter and non-winter death rates with 95% CIs in 1971–2019 using the age structure of the entire period as the standard population. Both absolute and relative differences between winter and non-winter rates were computed. To identify shifts in the trends, we explored the presence and location of breakpoints in the series of absolute and relative differences. We applied segmented linear regression analysis with one break on the cumulative values using the 'segmented' package[25] in R software V.4.2.2 (R Core Team).

To study the extent to which individual characteristics modified the risk of excess winter mortality, we focused on the more recent period from August 2000 to July 2019. We assessed effect modification using Poisson regression with interactions between the winter indicator and the covariates. These interaction effects, interpreted as relative risks, show how the effect of winter varies in multiplicative terms between different categories of modifying factors and the baseline reference group. Each potential effect modifier was studied separately, adjusting the

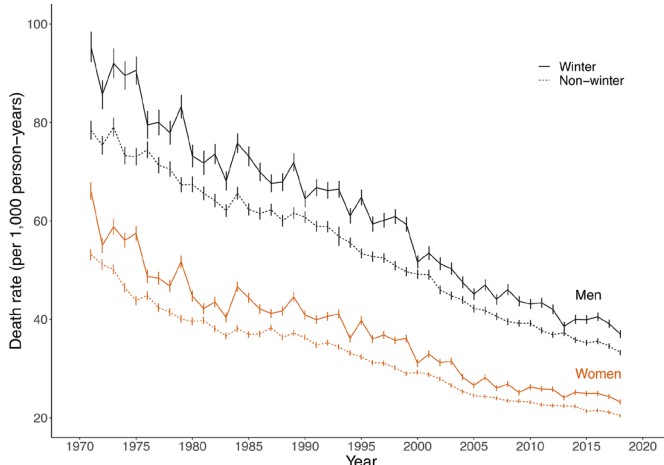

**Figure 1** All-cause winter and non-winter mortality rates, Finland 1971–2019, men and women aged 60 and over.

models for all the other covariates. To address potential issues of overdispersion in the Poisson regression models, robust, clustered SEs were employed, with each cluster defined as an individual.

## RESULTS
Over the study period from August 1971 to July 2019, 1 902 824 deaths were recorded, of which 36% occurred during winter.

The age-adjusted winter death rate fell from 95 deaths per 1000 person-years among men and 66 among women in 1971 to 37 and 23 in 2018, respectively (figure 1). The non-winter death rate was consistently below that of winter throughout the observation period and decreased from 78 to 33 among men and from 53 to 20 among women.

Absolute and relative differences between winter and non-winter mortality (figure 2) showed considerable year-to-year fluctuations. Overall, mortality declined over

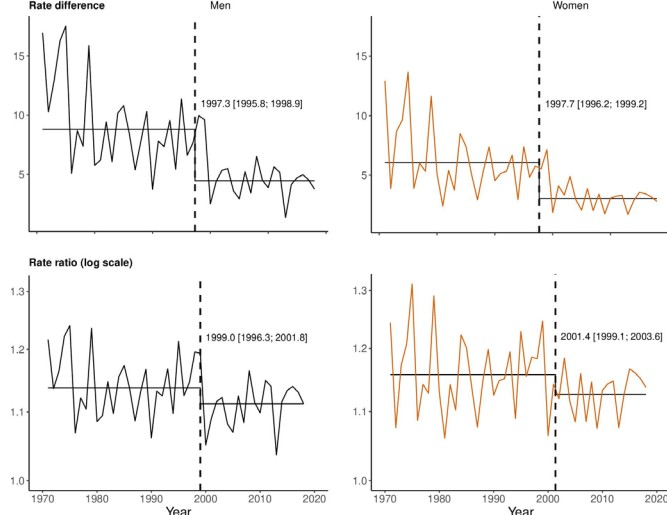

**Figure 2** Rate differences and rate ratios of winter and non-winter mortality rates with detected breakpoints (vertical dashed lines) and their 95% CIs, Finland 1971–2019, men and women aged 60 and over.

time, as did the absolute difference between winter and non-winter mortality rates—from an average of around 15 deaths per 1000 person-years to below five. Relative differences fluctuated between a winter excess of 31% (women year 1975) and only 4% (men year 2013), showing a clear downward trend. Changes in absolute differences for women were somewhat smaller than for men, and vice versa for relative differences, but peaks and troughs occurred in the same years for both sexes.

The values of both absolute and relative differences oscillated at lower levels towards the end of the study period, and the year-to-year variation became less pronounced. A breakpoint analysis indicated a change in the series around the year 2000 for both absolute and relative differences. The drop in the mean rate difference after the estimated break was nearly 50%, whereas for rate ratios (RRs), the drop was smaller.

Descriptive information about the study population at the end of the more recent period 2000–2019 shows population ageing and an increasing share of men (table 1). Most health conditions have become more common, except diseases of the circulatory system. The percentage of individuals recorded with dementia increased from 1.4% to 3.7%, reflecting the ageing population. However, the proportion with any recorded health condition during the previous calendar year remained at two-thirds. Living arrangements were stable, with around 3% living in institutions, one-third alone, and the rest with at least one other person.

In 2000–2019, all-cause mortality during winter occurred at an age-adjusted and sex-adjusted rate of 49.5 per 1000 person-years among men, and 31.4 per 1000 person-years among women, which was 11% and 14% higher than in the non-winter period, respectively (table 2). The winter excess varied by cause of death. It was observed for all cause of death groups, except for external causes among men, and was strongest for respiratory disease deaths. For both men and women, more deaths due to a respiratory cause occurred during the winter period (26% and 40% excess, respectively) than expected based on the non-winter average. Those who died from dementia showed the second-highest winter excess (men 21%, women 22%), exceeding that of circulatory disease deaths (men 13%, women 14%).

Table 3 shows age-adjusted winter and non-winter death rates and the winter/non-winter RRs for different categories of potential effect modifiers, adjusted for the other sociodemographic characteristics. For the winter/non-winter RRs, we also show the multiplicative interaction effects for each covariate, that is, the effect of winter relative to the ratio of the baseline reference group. The adjusted risk of dying in winter was slightly higher for women (1.14) than men (1.11), translating to a relative excess winter mortality that was 1.03 times higher among women than men (RR 1.03, 95% CI 1.02 to 1.04). The differences were clear between age groups with the most notable winter effects among the oldest-old, but there was no evidence that the risk of excess winter mortality varied

**Table 1** Changes in sociodemographic characteristics and pre-existing health conditions between 2000/2001 and 2018/2019, population aged 60 and over

| | 2000/2001 (%) | 2018/2019 (%) |
|---|---|---|
| Sociodemographic characteristics | | |
| Mean age (SD) | 71.2 (8.1) | 71.6 (8.4) |
| Sex | | |
| Men | 40.8 | 45.0 |
| Women | 59.2 | 55.0 |
| Age category | | |
| 60–69 | 47.2 | 46.8 |
| 70–79 | 35.9 | 34.0 |
| 80–89 | 14.7 | 15.9 |
| 90 and over | 2.1 | 3.2 |
| Income quartile | | |
| 1 (lowest) | 25.0 | 25.0 |
| 2 | 25.0 | 25.0 |
| 3 | 25.0 | 25.0 |
| 4 (highest) | 25.0 | 25.0 |
| Living arrangement | | |
| Household of >1 | 63.3 | 63.2 |
| Alone | 33.4 | 33.8 |
| Institution | 3.3 | 3.0 |
| Large regions | | |
| Coast | 36.4 | 38.7 |
| South | 29.0 | 27.8 |
| Centre | 19.2 | 17.9 |
| North | 15.5 | 15.6 |
| Pre-existing health conditions | | |
| Dementia | 1.4 | 3.7 |
| Mental health disorders | 3.7 | 3.8 |
| Cancer | 4.3 | 7.5 |
| Diseases of the nervous system | 4.4 | 6.7 |
| Diabetes | 7.4 | 15.0 |
| Diseases of the circulatory system | 44.4 | 35.5 |
| Diseases of the respiratory system | 9.4 | 11.4 |
| Alcohol-related causes | 0.3 | 0.5 |
| Accidents and violence | 4.4 | 6.4 |
| Other than specified above | 27.8 | 33.4 |
| Any health condition | 65.9 | 67.2 |
| Total | 100.0 | 100.0 |
| N | 1 035 643 | 1 559 165 |

by income and only weak evidence of differences between regions. The risk of dying in winter was relatively higher for persons living in institutions (RR 1.07, 95% CI 1.05 to 1.08), but those who lived alone did not differ much from those living with at least one other person. Overall, simultaneous adjustment for all covariates had very minor

**Table 2** Age-adjusted and sex-adjusted death rates in winter and non-winter periods and rate ratios, by cause of death and sex

| | Men | | | | Women | | | |
|---|---|---|---|---|---|---|---|---|
| | | Rate per 1000 person years | | Winter–non-winter rate ratio | | Rate per 1000 person years | | Winter–non-winter rate ratio |
| | Deaths | Winter | Non-winter | | Deaths | Winter | Non-winter | |
| Circulatory | 160 087 | 21.7 | 19.3 | 1.13 (1.12, 1.14) | 190 933 | 13.5 | 11.8 | 1.14 (1.13, 1.15) |
| Respiratory | 24 943 | 3.5 | 2.8 | 1.26 (1.23, 1.30) | 18 236 | 1.5 | 1.1 | 1.40 (1.35, 1.44) |
| Dementia | 36 659 | 6.0 | 4.9 | 1.21 (1.18, 1.24) | 82 108 | 5.7 | 4.7 | 1.22 (1.20, 1.24) |
| Other internal | 132 285 | 14.9 | 14.3 | 1.04 (1.03, 1.06) | 131 892 | 9.4 | 8.9 | 1.06 (1.05, 1.07) |
| External | 30 191 | 3.3 | 3.2 | 1.01 (0.99, 1.04) | 17 590 | 1.3 | 1.2 | 1.06 (1.03, 1.10) |
| All causes | 384 165 | 49.5 | 44.8 | 1.11 (1.10, 1.11) | 440 759 | 31.4 | 27.6 | 1.14 (1.13, 1.14) |

Finland 2000–2019, aged 60 and over.

effects on the results (results from unadjusted models not shown).

All analysed categories of pre-existing health conditions were associated with higher risk of death during both winter and non-winter periods. However, the multiplicative interaction effects of winter remained modest for nearly all health conditions. The most substantial excess mortality in winter was found for persons with a history of dementia with an adjusted winter/non-winter RR of 1.17 (95% CI 1.16 to 1.19) and a winter effect that was 1.06 times higher than among those without dementia (RR 1.06, 95% CI 1.04 to 1.07). Other categories of health conditions that seemed to slightly increase the risk of dying in winter included diseases of the respiratory system (RR 1.02, 95% CI 1.01 to 1.03) and mental disorders (RR 1.02, 95% CI 1.00 to 1.04). For alcohol-related causes and those with cancer, the effect was the other way around with relative risks of 0.93 (95% CI 0.89 to 0.98) and 0.93 (95% CI 0.92, 0.94), respectively.

Additional age-category-specific results (online supplemental tables S2 and S3) showed that while excess winter mortality increased with age, indications of effect modification by the analysed sociodemographic and health characteristics were mostly similar. Differences in the effect of winter were, however, more apparent among the younger age categories, whereas among 90 years, no effect modification was observed.

## DISCUSSION
### Main findings
With individual-level total population data covering nearly half a century, we examined long-term trends in excess winter mortality in Finland and assessed how individual-level characteristics may modify the risk of dying in winter. We showed that both winter and non-winter death rates steadily declined in 1971–2019, but winter rates remained above non-winter rates through the period. All-cause mortality was 11% and 14% higher during winter compared with non-winter months in 2000–2019 for men and women, respectively, with the highest

excess in respiratory and dementia-related causes of death. We found modest effect modification by sociodemographic characteristics and pre-existing health conditions. Advanced age, living in institutions and dementia increased the risk of winter mortality, while income level and most health conditions made no difference.

### Interpretation of the results
Previous research has documented downward trends in excess winter mortality during most of the 20th century.[2 3 26 27] The declining trend has been attributed to general socioeconomic progress, improved housing conditions and healthcare systems. According to previous research, differences between winter and summer mortality started to decline relatively early in Finland, in the 1930s,[28] and a slowly decreasing trend was also reported during the latter part of the century, until 1995.[29] Although not directly comparable to ours because of different definition of the winter excess (deaths in September–March compared with the month of lowest mortality) and age ranges considered, these previous results do align with our findings. In addition, with our results extending over 20 years further, we report a clear attenuation in the year-to-year variability during the last two decades and a downturn in the RRs around the year 2000.

Influenza-like illnesses are an important contributor to excess winter mortality, some studies showing that it is especially these—and not cold severity—that nowadays explains the year-to-year variations in the winter excess.[30 31] There are at least two influenza-related factors that may have contributed to the smaller winter mortality fluctuations observed towards the end of the study period. During the first decade of the 21st century, seasonal influenza activity and related mortality were relatively low compared with previous decades.[32] Also, since 2002, influenza vaccines have been administered as part of the national vaccination programme in Finland for all those aged 65 years and older. Vaccination is considered the best available intervention to prevent influenza illness, and some previous studies have reported that

Table 3   Death rates in winter and non-winter periods, rate ratios and relative risks of excess winter mortality by individual characteristics

| | Age adjusted rate per 1000 person years | | | |
| --- | --- | --- | --- | --- |
| | Winter (n=295 236) | Non-winter (n=529 688) | Full adjusted winter-non-winter rate ratio (95% CI)* | Full adjusted relative risk (RR) (95% CI)*† |
| **Sociodemographic characteristics** | | | | |
| All | 37.8 | 33.7 | 1.12 (1.12 to 1.13) | |
| Sex | | | | |
| Men | 49.5 | 44.8 | 1.11 (1.10 to 1.11) | 1.00 |
| Women | 31.4 | 27.6 | 1.14 (1.13 to 1.14) | **1.03 (1.02 to 1.04**) |
| Age category | | | | |
| 60–69 | 11.9 | 11.2 | 1.06 (1.05 to 1.07) | 1.00 |
| 70–79 | 30.7 | 27.9 | 1.10 (1.09 to 1.11) | **1.04 (1.02 to 1.05**) |
| 80–89 | 94.6 | 83.2 | 1.14 (1.13 to 1.15) | **1.07 (1.06 to 1.09**) |
| 90 and over | 269.8 | 228.0 | 1.19 (1.17 to 1.20) | **1.12 (1.10 to 1.13**) |
| Income quartile | | | | |
| 1 (lowest) | 41.5 | 37.2 | 1.12 (1.11 to 1.13) | 1.00 |
| 2 | 38.5 | 34.3 | 1.12 (1.11 to 1.13) | 1.00 (0.99 to 1.02) |
| 3 | 37.9 | 33.7 | 1.12 (1.11 to 1.13) | 1.01 (0.99 to 1.02) |
| 4 (highest) | 32.9 | 29.2 | 1.13 (1.11 to 1.14) | 1.01 (0.99 to 1.02) |
| Living arrangement | | | | |
| Household of >1 | 31.0 | 28.0 | 1.10 (1.10 to 1.11) | 1.00 |
| Alone | 34.3 | 30.6 | 1.12 (1.11 to 1.13) | **1.01 (1.00 to 1.02**) |
| Institution | 110.1 | 93.6 | 1.18 (1.16 to 1.19) | **1.07 (1.05 to 1.08**) |
| Region | | | | |
| Coast | 36.8 | 33.1 | 1.11 (1.10 to 1.12) | 1.00 |
| South | 37.5 | 33.2 | 1.13 (1.12 to 1.14) | **1.02 (1.00 to 1.03**) |
| Centre | 38.9 | 34.8 | 1.12 (1.11 to 1.13) | 1.01 (0.99 to 1.02) |
| North | 39.3 | 34.7 | 1.13 (1.12 to 1.15) | **1.02 (1.01 to 1.03**) |
| **Pre-existing health conditions** | | | | |
| Dementia | | | | |
| No | 34.0 | 30.6 | 1.11 (1.11 to 1.12) | 1.00 |
| Yes | 78.9 | 67.2 | 1.17 (1.16 to 1.19) | **1.06 (1.04 to 1.07**) |
| Mental health disorders | | | | |
| No | 36.5 | 32.6 | 1.12 (1.12 to 1.13) | 1.00 |
| Yes | 69.6 | 60.9 | 1.14 (1.12 to 1.16) | **1.02 (1.00 to 1.04**) |
| Cancer | | | | |
| No | 34.9 | 30.7 | 1.14 (1.13 to 1.14) | 1.00 |
| Yes | 74.1 | 70.4 | 1.05 (1.04 to 1.07) | **0.93 (0.92 to 0.94**) |
| Diseases of the nervous system | | | | |
| No | 36.1 | 32.2 | 1.12 (1.12 to 1.13) | 1.00 |
| Yes | 65.6 | 58.7 | 1.12 (1.10 to 1.13) | 0.99 (0.98 to 1.01) |
| Diabetes | | | | |
| No | 35.2 | 31.4 | 1.12 (1.12 to 1.13) | 1.00 |
| Yes | 57.0 | 50.5 | 1.13 (1.12 to 1.14) | 1.01 (1.00 to 1.02) |
| Circulatory | | | | |
| No | 29.9 | 26.8 | 1.12 (1.11 to 1.13) | 1.00 |

**Table 3** Continued

| | Age adjusted rate per 1000 person years | | | |
| | Winter (n=295 236) | Non-winter (n=529 688) | Full adjusted winter-non-winter rate ratio (95% CI)* | Full adjusted relative risk (RR) (95% CI)*† |
| --- | --- | --- | --- | --- |
| Yes | 45.5 | 40.4 | 1.13 (1.12 to 1.13) | 1.01 (1.00 to 1.02) |
| Respiratory | | | | |
| No | 34.7 | 31.0 | 1.12 (1.11 to 1.12) | 1.00 |
| Yes | 61.2 | 53.6 | 1.14 (1.13 to 1.15) | **1.02 (1.01 to 1.03)** |
| Alcohol related | | | | |
| No | 37.5 | 33.4 | 1.12 (1.12 to 1.13) | 1.00 |
| Yes | 169.7 | 161.8 | 1.05 (1.00 to 1.10) | **0.93 (0.89 to 0.98)** |
| Accidents and violence | | | | |
| No | 36.4 | 32.5 | 1.12 (1.12 to 1.13) | 1.00 |
| Yes | 56.0 | 49.7 | 1.13 (1.11 to 1.14) | 1.00 (0.99 to 1.02) |
| Other than specified above | | | | |
| No | 34.1 | 30.1 | 1.13 (1.13 to 1.14) | 1.00 |
| Yes | 44.2 | 39.9 | 1.11 (1.10 to 1.12) | **0.98 (0.97 to 0.99)** |
| Any health condition | | | | |
| No | 19.2 | 17.2 | 1.12 (1.11 to 1.13) | 1.00 |
| Yes | 43.7 | 38.9 | 1.12 (1.12 to 1.13) | 1.00 (0.99 to 1.02) |

Finland 2000–2019, men and women aged 60 and over.
Bold values indicate estimates with 95% CIs that do not contain 1.
*Models are adjusted for sex, age, income, living arrangement, region and year.
†RRs in the last column are the interaction terms of the winter indicator and the effect modifier. These show the relative differences in the effect of winter on mortality between effect modifier categories.

the influenza vaccine significantly reduces mortality.[33 34] Although more research is needed as the mortality benefits of influenza vaccine, especially among the older population, have been questioned,[35 36] our results are consistent with the idea that both lower influenza activity of recent decades and the introduction of vaccinations may have reduced excess winter mortality peaks.

Age is known to play an important role for excess winter mortality,[12 37] and of all the individual-level effect modifiers included in our analysis, older age showed the strongest and clearest influence. The increasing risk with age is commonly interpreted as a reflection of increasing frailty and vulnerability towards adverse environmental stresses as age advances.[38] Accordingly, we found that persons living in institutions were likelier to die in winter. As previous studies with similar findings from France,[16] Netherlands[17] and England and Wales[18] note, those living in institutions are rarely exposed to outside weather and cold. However, they are likely to spend more time indoors within confined environments, which may increase the spread of infections.[39] Furthermore, the prevalence of frailty is considerably high among older adults living in residential care,[40] indicating their increased risk of mortality and vulnerability to external stressors.[41]

Apart from frailty associated with advanced age, it is generally assumed that excess winter mortality most importantly affects persons with pre-existing health conditions. Nevertheless, previous literature approaching the topic by including information about the prior health of those dying in winter is scarce. In general, our findings showed very weak or no evidence for increased risk for dying in winter among those with pre-existing conditions. Only persons with dementia emerged as being at greater risk, a finding in line with few previous studies. Our results concerning dementia mortality also support these findings, as those who died from dementia showed the second-highest winter excess of around 21%. Previously, Liddell et al[42] have also reported almost a third more dementia deaths in winter compared with non-winter times in the UK. While a tendency to attribute deaths to the underlying chronic condition instead of a respiratory infection could explain some of the excesses, another possible interpretation is that those with dementia have difficulties in communicating changes in their health status to relatives and care staff. In a study on susceptibility to cold, Zanobetti et al[43] found that in the USA, persons with previous admission for dementia were at greater risk of dying on extremely cold days and it has been speculated that this may relate to reduced perception of and protection from cold.[44] Living in institutions, as discussed above, has also been suggested to play a role. A recent study[45] on COVID-19 pandemic-related excess

mortality in the UK proposed that the highest relative increases of death for individuals with dementia may be attributed to higher exposure to seasonal respiratory viruses due to their more frequent residency in institutional settings. However, supplementary analyses (online supplemental table S4) in this study showed that effect modification of winter mortality by age or pre-existing diagnosis of dementia was in fact modest or non-existent among those living in institutions. These findings suggest that the elevated excess winter mortality in individuals with pre-existing dementia cannot be solely attributed to institutional living.

The risk of dying in winter was not significantly increased for those with other pre-existing health conditions. Not even those with respiratory or circulatory conditions seemed significantly more vulnerable, although persons with these conditions are traditionally assumed to face a particularly high risk of death in winter. This common assumption may partly arise from considering stated causes of death on the death certificate as evidence of pre-existing conditions and thus interpreting the findings that show high excess winter mortality for cardiovascular and respiratory causes as an indication that individuals with pre-existing cardiovascular or respiratory disease are more susceptible to the adverse effects of winter. We also found the highest excess winter mortality in deaths due to diseases of the respiratory system and a somewhat lesser excess risk in deaths due to circulatory diseases, but a similar excess risk did not apply to people diagnosed with these conditions in the preceding year. Previous evidence on the effect of pre-existing health conditions—primarily based on studies on cold-related mortality and not on excess winter mortality specifically—is conflicting. For example, while Wilkinson *et al*[22] found that pre-existing respiratory disease increased the risk of a winter death, and Schwartz[46] reported elevated risks on cold days for persons with chronic obstructive pulmonary disease, other empirical studies have found no higher risks than average[47] and even lower than average risks have been reported.[43] Similarly, for cardiovascular conditions, there is some evidence both for[48 49] and against[46 50 51] an increased risk of winter death. One possible explanation for not detecting higher risks for most of the pre-existing health conditions may be that persons with a diagnosis and previous healthcare contacts are the ones who receive the most benefit from preventive measures. If true, this suggests potential for prevention among the apparently healthy population.

While there is a well-known link between socioeconomic position and overall mortality,[52] we found no evidence to suggest a socioeconomic gradient in vulnerability to excess winter mortality when measured by income. Instead, the increase in mortality in winter was similar in all income categories. Previous studies regarding the association between socioeconomic factors and excess winter mortality have provided conflicting results.[13] Hales *et al*,[53] for example, found that the risk of winter death in New Zealand was higher for those in the lowest income

tertile compared with those in the highest tertile. Many of the proposed mechanisms through which lower socioeconomic position and income may increase the risk of winter death are linked to housing conditions and factors such as central heating and fuel poverty.[13] However, it has also been noted that low socioeconomic position does not always go hand in hand with the poorest quality housing, as in some countries, or example, the UK, social housing may even have better than average energy efficiency characteristics.[22] In Finland, housing is generally of high quality, and conditions are relatively good at all income levels. Homes and buildings are well insulated, and most households can keep their homes adequately warm regardless of low outside temperatures,[54] which probably partly explains why no income gradient was observed in our study. Similar conclusions were drawn from a previous study in another Nordic country, Denmark.[12] Future studies should, however, assess possible interactions between income and living arrangements in diverse social settings to gain a more comprehensive understanding of their effects on excess winter mortality.

### Methodological considerations

Our study is based on full-coverage high-quality, population-level register data with minimal attrition and no underreporting or recall bias. The use of register data allowed us to examine risk factors at the individual instead of aggregate level, overcoming a key limitation of most previous work on excess winter mortality. Nevertheless, we recognise that identifying persons with pre-existing health conditions based on healthcare visits in specialised care and reimbursement for medicine costs may present some limitations as these records may partly reflect differences in health-seeking behaviour and ignore less severe health problems. Another potential cause of concern relates to the timing of the measurement. If we measured health too close to death, we might be capturing the beneficial effect of being monitored by healthcare professionals or the results could be affected by a survivor bias. Our main results were based on measuring health conditions during the preceding calendar year (a 7-month 'quarantine' period from January to July). We replicated the analysis measuring health closer (no quarantine period before start of follow-up) and further back (quarantine period extended to 2 years). Results, however, remained unchanged, indicating that the timing of the measurement is not driving the small effects.

Our analyses on winter excess mortality in different causes of death were based solely on the underlying cause of death. Thus, our findings may significantly underestimate the total impact of respiratory infections and influenza-like illnesses on excess winter mortality.[21] Deaths are rarely coded as influenza in death certificates[55] but instead attributed to chronic diseases, especially among the older population. Nevertheless, respiratory infections are important triggers of, for instance, cardiovascular problems.[56] To better understand the mechanisms behind excess winter mortality, future research should

consider information on contributory causes of death to assess the indirect triggering effect of respiratory diseases.

## CONCLUSIONS AND IMPLICATIONS

In conclusion, with high-quality individual-level total population data, we showed that winter mortality has declined since 1971, and particularly after year 2000, but the winter mortality rates have remained above non-winter rates throughout the past five decades. We provide robust evidence that those with advanced age, living in institutions and dementia had a particularly high risk of winter mortality. The subgroup differences were relatively small overall, suggesting an increased risk of winter death even beyond the more vulnerable subgroups. In the context of population ageing and increasing numbers of individuals surviving into ages where excess winter mortality reaches high levels, public health action to protect high-risk groups may have major population-level benefits.

**Author affiliations**
[1]Helsinki Institute for Demography and Population Health, University of Helsinki Faculty of Social Sciences, Helsinki, Finland
[2]Max Planck – University of Helsinki Center for Social Inequalities in Population Health, Helsinki, Finland
[3]International Max Planck Research School for Population, Health and Data Science, Rostock, Germany
[4]The London School of Economics and Political Science Department of Social Policy, London, UK
[5]Max-Planck-Institute for Demographic Research, Rostock, Germany

**Acknowledgements** US gratefully acknowledges the resources provided by the International Max Planck Research School for Population, Health and Data Science (IMPRS-PHDS).

**Contributors** US, HR, LT, MM and PM contributed to the conception and design of the study. PM acquired the data. US conducted the literature review, prepared the data, conducted the analyses and wrote the first version of the manuscript. All authors critically revised, commented and edited the draft versions of the manuscript for important intellectual content. All authors have approved the final version to be published and agree to be accountable for all aspects of the work. US is responsible for the overall content as guarantor.

**Funding** US was supported by Jenny and Antti Wihuri Foundation (grant numbers 00210374 and 00220382). PM was supported by the European Research Council under the European Union's Horizon 2020 research and innovation programme (grant agreement number 101019329), the Strategic Research Council (SRC) within the Academy of Finland grants for ACElife (352543-352572) and LIFECON (308247), and grants to the Max Planck–University of Helsinki Center from the Jane and Aatos Erkko Foundation (210046), the Max Planck Society (5714240218), University of Helsinki (77204227) and Cities of Helsinki, Vantaa and Espoo.

**Disclaimer** The study does not necessarily reflect the Commission's views and in no way anticipates the Commission's future policy in this area. The funders had no role in the study design, data collection and analysis, decision to publish, or preparation of the manuscript.

**Competing interests** None declared.

**Patient and public involvement** Patients and/or the public were not involved in the design, or conduct, or reporting, or dissemination plans of this research.

**Patient consent for publication** Not applicable.

**Ethics approval** The use of the anonymised register data has been approved by the ethics boards of Statistics Finland and FinData (permissions TK-53-1490-18 and THL/2180/14.02.00/2020). The register data are originally collected for administrative and statistical purposes. No informed consent is required for register-based studies in Finland.

**Provenance and peer review** Not commissioned; externally peer reviewed.

**Data availability statement** Data may be obtained from a third party and are not publicly available. Due to data protection regulations of the register holders, the authors are unable to make the data available to third parties. In order to use the data, researchers and research institutions need to apply for a licence that authorises its use for research purposes. Access to the data can be requested by contacting the register holders.

**ORCID iD**
Ulla Suulamo http://orcid.org/0000-0002-5417-3236

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
