## [Reviewer comments · BMJ Open]

ARTICLE DETAILS

TITLE (PROVISIONAL)	Excess winter mortality in Finland in 1971-2019 – a register-based study on long-term trends and effect modification by sociodemographic characteristics and pre-existing health conditions
AUTHORS	Suulamo, Ulla; Remes, Hanna; Tarkiainen, Lasse; Murphy, Michael; Martikainen, Pekka

VERSION 1 – REVIEW

REVIEWER	Jones, Rodney Healthcare Analysis & Forecasting
REVIEW RETURNED	10-Sep-2023

GENERAL COMMENTS	Dear authors, Congratulations on a valuable paper. I cannot fault your data analysis and its conclusions. However, I have published several relevant studies which are not referenced. I am aware of an Irish study specifically mentioning the word dementia which is relevant. A group from New Zealand has just published a preprint The rise and fall of EWM in New Zealand https://www.researchsquare.com/article/rs-3204677/v1. The papers of Peter Doshi regarding influenza and influenza vaccination should be cited and indeed I have also published a paper questioning the consistent effects of influenza vaccination against all-cause winter mortality. You may like to include a section in your analysis looking at the proportion aged 65+ who are vaccinated in Finland over time verses the effect on all-cause mortality. In conclusion, a good study which can be elevated to an even better study by some additional research and a deeper delve into the literature.
--

REVIEWER	Leon, David London School of Hygiene and Tropical Medicine
REVIEW RETURNED	19-Sep-2023

GENERAL COMMENTS	This is an elegant and well written paper that addresses a longstanding issue : that of excess winter deaths. It uses a linkage of routine mortality, administrative and health data at the individual level for the whole Finnish population over a 50 year period, with a focus on deaths 2000-2019. The results are consistent with those of previous work suggesting that in high income countries excess winter mortality (relative to the non-winter period) is declining in absolute terms over time, and since 2000 appears to have declined in relative terms as well. What is striking is that the author's show that there is little evidence that income or region moderate the
--

	relative winter/non-winter excess. However, age, sex and living in an institution are significant effect modifiers. In terms of pre-existing disease dementia stands out as showing the largest modifying effect, with those with such a diagnosis showing a bigger winter/non-winter excess relative to those without such a diagnosis. It is striking that a pre-existing diagnosis of respiratory disease shows a much smaller effect i.e. there is little difference in the relative seasonal excess mortality among those with respiratory diagnosis vs without. The authors discuss their findings clearly and place them in the context of the broader literature, and identify several weaknesses of their study. Overall, this paper provides a valuable addition to the literature. Looking in particular at the potential modifying effect of pre-existing health conditions echoes many papers that have been published during and since the pandemic that have examined whether pre-existing disease resulted in higher COVID-related mortality (or overall excess deaths). Indeed there are some interesting parallels in their findings with a study we published on COVID excess mortality in the UK (https://pubmed.ncbi.nlm.nih.gov/34990450/) in which we found those with pre-existing dementia (and also learning difficulties) showed strikingly large relative increases in all cause mortality in the first wave of the pandemic, whereas other pre-existing conditions did not. We interpreted this as suggesting that people with dementia (and severe learning difficulties) were much more likely to live in institutions than those without, and as a result were much more likely to be exposed to COVID and die from it. The authors of the current paper suggest a similar potential effect with respect to the higher likelihood of being exposed to seasonal influenza virus if you live in an institution. I would suggest that the authors elaborate on this possibility more in their analyses and discussion as outlined below :  1. In Table 3 the winter-nonwinter interaction has been adjusted for a variety of different factors including living condition, where living in an institution is one of the 3 categories. I would like to see a table in which sex, age, dementia and respiratory disease interaction effects are shown separately stratified by living in an institution or not. This would allow the reader to get a better sense of whether the age and dementia effects in particular are very different in those in institutions vs those not in institutions. At the moment we cannot see this. I also have some more minor suggestions :  2. Figure 2 should show the rate ratios on a log scale 3. More information should be given about the sensitivity analyses described on lines 12-17 page 11 where the time window for pre-existing disease is varied in width. There is a slight concern that people with dementia may not have an active consultation in the 1 year window and thus be misclassified. 4. Why did the authors not include cancer as a pre-existing condition? If possible I suggest this is added in. 5. Please explain how household income is determined for people living in institutions.
--	--

REVIEWER	Wong, Esther University of Bristol
REVIEW RETURNED	22-Sep-2023

GENERAL COMMENTS	Ulla Suulamo's research aimed to evaluate the trends in winter excess mortality over the past few decades using population-level registry data. Several analyses also took into consideration
---

	sociodemographic characteristics and pre-existing health conditions, to identify vulnerable groups and assess changes in the risk trend over time, with the ultimate goal of implementing appropriate preventive measures. While the work is interesting, there are a few comments and suggestions to enhance the clarity and completeness of the study:  1) In Table 1, instead of "Characteristics of study populations in 2000/01 and 2018/19," it is recommended to use a title that succinctly describes the objective of the table, such as "Changes in Sociodemographic Characteristics and Pre-existing Health Conditions between 2000/01 and 2018/19." 2) In Table 1, the significant increase in reported cases of dementia between 2000/01 and 2018/19 was noted. Given the aging population during this period, it is prudent to consider age as a contributing factor to the rise in dementia. Furthermore, it's worth discussing the possible correlation between dementia and residing in institutions, as individuals with dementia might be more likely to live in such settings. 3) In the methods section, it is stated that "Household disposable income was categorized into quartiles in the population aged 60 and over and adjusted for the household composition." It was assumed that this categorization was applied to the entire period from 2000 to 2019. However, it is surprising to observe that the income quartile structure remained unchanged from 2000/01 to 2018/19, with all values at 25.0. To provide a clearer perspective on the income distribution within the study population, an alternative approach could be to use the income quartiles of the entire Finnish population (all age groups) to demonstrate the income distribution within the study population relative to the national figures. 4) It would be valuable to explore potential interactions between income and living arrangements, or between living arrangements and age, to gain a more comprehensive understanding of their effects on the study outcomes. 5) In Table 3, there is an asterisk after "Rate per 1000 person years*.", if there are any additional information or context related to the rates, it is advisable to provide an explanation for this asterisk in the footnote. 6) Table 3 presents the winter-nonwinter rate ratio in two models: one adjusted only for age (superscript a) and another adjusted for sex, age, income, living arrangement, region, and year (superscript b). A rationale for this distinction should be provided to justify why age alone was considered in the first model while other sociodemographic characteristics were added in the second model. 7) Given the involvement of multiple analyses with different cohorts, it is recommended to include a Consort diagram or a table that delineates the various cohorts used in the relevant analyses. This will help the reader better understand the study's design and population selection. 8) Table S1 mentions the exclusion of alcohol-related conditions within mental health disorders, circulatory issues, and accidents and violence. To enhance transparency, it is advisable to detail this exclusion process in the methods section, and indicate whether any analyses of interactions within the pre-existing health conditions were conducted. 9) The authors might consider assessing multi-morbidities of individuals rather than solely relying on dichotomous pre-existing health conditions. Alternatively, they should provide a justification
--	---

	for not considering multi-morbidities if that was the chosen approach. 10) In the analysis using multivariable Poisson regression to assess the risk of excess winter mortality, it is important to check for overdispersion in the model and report the findings accordingly. 11) From the abstract and title, it has been stated that this is a prospective nationwide register study. However, in the methods section, it was mentioned, "This data was drawn from censuses and population registers..." and "All data was collected for routine administrative purposes..." If the data had already been collected before the initiation of the research, or were collected for other purposes but not specifically for the study, it would be more accurate to classify as a retrospective study. Therefore, please reconsider clarifying the study design type to ensure accuracy.
--	---

REVIEWER	Dahl, Cecilie University of Oslo, Department of Community Medicine and Global health
REVIEW RETURNED	24-Sep-2023

GENERAL COMMENTS	Dear Authors, Thank you for providing this exciting and relevant paper for review. I would like to congratulate you. This is well-written paper on a topic that has not been much studied, and the results are important to elucidate the role of the cold climate in health and disease in the Nordic countries. It also takes advantage of already existing data for a very long time period, showing how important societal changes are in mitigating the effects from this harsh climate. I really enjoyed reading it, and my comments below, except for the last one about the interactions are only minor (to provide further clarity).  - First: An ethics statement is needed. I understand that consent is impossible, but did you obtain exception from consent for the registries? Did you need ethical consent from a committee for the linkages of the Census and the registries. - "Data" are plural. Therefore "data were" not "data was" (please change throughout the manuscript) - Table 3: are some category headings missing above yes and no at the end? - How did you obtain the estimates for the effect modifications/interactions? Some more information is required under "statistics" in the method section. Usually, the interactions are tested (based on pre-defined hypotheses), then estimates are obtained by either stratifying on the variables of interest or predicted for each level of the variables (i.e., obtaining marginal effects) and lastly compared to obtain effect estimates. Note: the estimates that you obtain from the statistical interaction itself, or the model including the interaction term are not valid.
--

VERSION 1 – AUTHOR RESPONSE

Reviewer Reports:

Reviewer: 1

Dr. Rodney Jones, Healthcare Analysis & Forecasting

Comments to the Author:

Dear authors,

Congratulations on a valuable paper. I cannot fault your data analysis and its conclusions. However, I have published several relevant studies which are not referenced. I am aware of an Irish study specifically mentioning the word dementia which is relevant. A group from New Zealand has just published a preprint The rise and fall of EWM in New Zealand

<https://www.researchsquare.com/article/rs-3204677/v1>. The papers of Peter Doshi regarding influenza and influenza vaccination should be cited and indeed I have also published a paper questioning the consistent effects of influenza vaccination against all-cause winter mortality. You may like to include a section in your analysis looking at the proportion aged 65+ who are vaccinated in Finland over time verses the effect on all-cause mortality.

In conclusion, a good study which can be elevated to an even better study by some additional research and a deeper delve into the literature.

Reviewer: 2

Prof. David Leon, London School of Hygiene and Tropical Medicine

Comments to the Author:

This is an elegant and well written paper that addresses a longstanding issue : that of excess winter deaths. It uses a linkage of routine mortality, administrative and health data at the individual level for the whole Finnish population over a 50 year period, with a focus on deaths 2000-2019. The results are consistent with those of previous work suggesting that in high income countries excess winter mortality (relative to the non-winter period) is declining in absolute terms over time, and since 2000 appears to have declined in relative terms as well. What is striking is that the author's show that there is little evidence that income or region moderate the relative winter/non-winter excess. However, age, sex and living in an institution are significant effect modifiers. In terms of pre-existing disease dementia stands out as showing the largest modifying effect, with those with such a diagnosis showing a bigger winter/non-winter excess relative to those without such a diagnosis. It is striking that a pre-existing diagnosis of respiratory disease shows a much smaller effect i.e. there is little difference in the relative seasonal excess mortality among those with respiratory diagnosis vs without.

The authors discuss their findings clearly and place them in the context of the broader literature, and identify several weaknesses of their study. Overall, this paper provides a valuable addition to the literature. Looking in particular at the potential modifying effect of pre-existing health conditions echoes many papers that have been published during and since the pandemic that have examined whether pre-existing disease resulted in higher COVID-related mortality (or overall excess deaths). Indeed there are some interesting parallels in their findings with a study we published on COVID excess mortality in the UK (<https://pubmed.ncbi.nlm.nih.gov/34990450/>) in which we found those with pre-existing dementia (and also learning difficulties) showed strikingly large relative increases in all cause mortality in the first wave of the pandemic, whereas other pre-existing conditions did not. We

interpreted this as suggesting that people with dementia (and severe learning difficulties) were much more likely to live in institutions than those without, and as a result were much more likely to be exposed to COVID and die from it. The authors of the current paper suggest a similar potential effect with respect to the higher likelihood of being exposed to seasonal influenza virus if you live in an institution. I would suggest that the authors elaborate on this possibility more in their analyses and discussion as outlined below:

1. In Table 3 the winter-nonwinter interaction has been adjusted for a variety of different factors including living condition, where living in an institution is one of the 3 categories. I would like to see a table in which sex, age, dementia and respiratory disease interaction effects are shown separately stratified by living in an institution or not. This would allow the reader to get a better sense of whether the age and dementia effects in particular are very different in those in institutions vs those not in institutions. At the moment we cannot see this.

Response: Thank you for raising the concern regarding the potential role of living arrangements in excess winter mortality among individuals with pre-existing dementia. We agree that this is an important point. We have conducted additional analyses and added a new supplementary table “Table S4. Relative risks of excess winter mortality by selected individual characteristics, whether living in institutions or community. Finland 2000-2019, men and women aged 60 and over.” following the suggestion.

Older individuals, and particularly those with dementia, are more likely to be residing in institutional settings – in our study population one quarter of those having a dementia diagnosis lived in institutions. This higher likelihood of institutional residence may expose these individuals to the spread of respiratory infections, potentially explaining their elevated relative excess mortality during winter compared to those without dementia. However, when we stratified the analysis by living in an institution or not, we found that among those in institutions, the effect modification was modest, and winter appeared to increase the mortality risk across most subgroups to a similar degree. Conversely, individuals with dementia living in the community were at an increased risk of mortality in winter compared to those without dementia. These findings suggest that the heightened excess winter mortality in individuals with pre-existing dementia cannot be solely attributed to institutional living. It is possible that although those living in institutions may face higher exposure, their closer monitoring could result in earlier and more effective treatment.

The following text has been added to the discussion section (page 12): “Living in institutions, as discussed above, has also been suggested to play a role. A recent study⁴⁵ on COVID-19 pandemic-related excess mortality in the UK proposed that the highest relative increases of death for individuals with dementia may be attributed to higher exposure to seasonal respiratory viruses due to their more frequent residency in institutional settings. However, supplementary analyses (Table S4) in this study showed that effect modification of winter mortality by age or pre-existing diagnosis of dementia was in fact modest or non-existent among those living in institutions. These findings suggest that the elevated excess winter mortality in individuals with pre-existing dementia cannot be solely attributed to institutional living.”

The following table has been added to the Supplementary Material:

Table S4. Relative risks of excess winter mortality by selected individual characteristics, whether living in institutions or community. Finland 2000-2019, men and women aged 60 and over.

	Winter-nonwinter interaction (95% CI)	
	In institution n=156,851	In community n=668,073
Winter-nonwinter rate ratio	1.18 (1.16, 1.19)	1.11 (1.10, 1.12)
Sociodemographic characteristics		
Sex		
Men	1.00	1.00
Women	1.01 (0.99, 1.03)	1.02 (1.01, 1.04)
Age category		
60–69	1.00	1.00
70–79	1.02 (0.97, 1.07)	1.03 (1.02, 1.05)
80–89	1.01 (0.96, 1.06)	1.07 (1.06, 1.09)
90 and over	1.05 (1.00, 1.10)	1.11 (1.09, 1.13)
Pre-existing health conditions		
Dementia		
No	1.00	1.00
Yes	1.00 (0.98, 1.02)	1.06 (1.05, 1.08)
Respiratory		
No	1.00	1.00
Yes	1.01 (0.98, 1.04)	1.03 (1.01, 1.04)

Note: Relative risks are the interaction terms of the winter indicator and the effect modifier. These show the relative differences in the effect of winter on mortality between effect modifier categories. Models are adjusted for sex, age, income, living arrangement, region, and year.

I also have some more minor suggestions:

2. Figure 2 should show the rate ratios on a log scale

Response: Thank you for the suggestion. We have now log scaled the rate ratios and the updated figures are available for review below as well as in the submission.

3. More information should be given about the sensitivity analyses described on lines 12-17 page 11 where the time window for pre-existing disease is varied in width. There is a slight concern that people with dementia may not have an active consultation in the 1-year window and thus be misclassified.

Response: Thank you for noting the need for more details on the measurement of health conditions. It is certainly true that all persons with a chronic condition may not have active consultation each year, but as our measure is also based on medication use, we are likely to capture the great majority of pre-existing chronic health conditions. In our data over two thirds of those identified with pre-existing dementia originate from the special reimbursement register which enables the identification of clinically diagnosed patients who are eligible for an elevated reimbursement for their medicine expenditure due to certain chronic conditions. The register holds information on starting and ending dates for the reimbursement right, but for diagnoses such as dementia the reimbursement right is usually valid indefinitely. This means that once a right is granted it will show in the register each year after that. We believe the proportion of individuals with diagnosed dementia who neither use medication nor specialized healthcare is likely to be too small to significantly bias our main conclusions.

4. Why did the authors not include cancer as a pre-existing condition? If possible, I suggest this is added in.

Response: Initially, cancer was excluded from the list of pre-existing conditions based on the common findings in previous studies, which indicated no winter excess for cancer deaths. However, in response to your suggestion, we have reconsidered and incorporated cancer into the list of pre-existing conditions. Our results align with the existing body of research on excess winter cancer mortality and show that individuals with a cancer diagnosis do not exhibit winter excess compared to those without cancer. All relevant sections of the manuscript including the tables and methods and results sections, have been revised accordingly to reflect this inclusion. The following pieces of text have been incorporated (pages 8-9): “For alcohol-related causes and those with cancer, the effect was the other way around with rate ratios of 0.93 (95% CI 0.89–0.98) and 0.93 (95% CI 0.92, 0.94), respectively.”

5. Please explain how household income is determined for people living in institutions.

Response: For persons living in institutions income is determined as personal disposable income. This is stated in the methods section as follows (page 4): “For persons living in institutions, we used personal disposable income.”

Reviewer: 3

Dr. Esther Wong, University of Bristol

Comments to the Author:

Ulla Suulamo's research aimed to evaluate the trends in winter excess mortality over the past few decades using population-level registry data. Several analyses also took into consideration sociodemographic characteristics and pre-existing health conditions, to identify vulnerable groups and assess changes in the risk trend over time, with the ultimate goal of implementing appropriate preventive measures. While the work is interesting, there are a few comments and suggestions to enhance the clarity and completeness of the study:

1) In Table 1, instead of "Characteristics of study populations in 2000/01 and 2018/19," it is recommended to use a title that succinctly describes the objective of the table, such as "Changes in Sociodemographic Characteristics and Pre-existing Health Conditions between 2000/01 and 2018/19."

Response: Thank you for this suggestion to make the objective of the table clearer. The title of Table 1 now reads “Changes in sociodemographic characteristics and pre-existing health conditions between 2000/01 and 2018/19, population aged 60 and over.” Likewise, the title of the Table S2 in the

Supplemental Material has been changed to: "Changes in sociodemographic characteristics and pre-existing health conditions between 2000 and 2018, by age category, population aged 60 and over."

2) In Table 1, the significant increase in reported cases of dementia between 2000/01 and 2018/19 was noted. Given the aging population during this period, it is prudent to consider age as a contributing factor to the rise in dementia. Furthermore, it's worth discussing the possible correlation between dementia and residing in institutions, as individuals with dementia might be more likely to live in such settings.

Response: Thank you for your insightful observation. We acknowledge the significant increase in reported cases of dementia as highlighted in Table 1. Considering the aging population during this period, age indeed plays a crucial role in the rise of dementia cases. To clarify this association, we have incorporated the following (page 6): "The percentage of individuals recorded with dementia increased from 1.4 to 3.7%, reflecting the ageing population."

In relation to the second point on the association between dementia, residing in institutions and excess winter mortality, to further elucidate this relationship we have conducted additional analyses and present these results in a new supplementary table S4 titled "Relative risks of excess winter mortality by selected individual characteristics, whether living in institutions or community. Finland 2000-2019, men and women aged 60 and over.". Individuals with dementia are more likely to reside in institutional settings and therefore this subgroup may be more exposed to the spread of respiratory infections, potentially explaining their larger relative excess mortality during winter compared to those without dementia. However, the results of Table S4 suggest that the increased excess winter mortality of those with pre-existing dementia was not explained by institutional living.

We have added the following to the manuscript (page 12): Living in institutions, as discussed above, has also been suggested to play a role. A recent study⁴⁵ on COVID-19 pandemic-related excess mortality in the UK proposed that the highest relative increases of death for individuals with dementia may be attributed to higher exposure to seasonal respiratory viruses due to their more frequent residency in institutional settings. However, supplementary analyses (Table S4) in this study showed that effect modification of winter mortality by age or pre-existing diagnosis of dementia was in fact modest or non-existent among those living in institutions. These findings suggest that the elevated excess winter mortality in individuals with pre-existing dementia cannot be solely attributed to institutional living.

3) In the methods section, it is stated that "Household disposable income was categorized into quartiles in the population aged 60 and over and adjusted for the household composition." It was assumed that this categorization was applied to the entire period from 2000 to 2019. However, it is surprising to observe that the income quartile structure remained unchanged from 2000/01 to 2018/19, with all values at 25.0. To provide a clearer perspective on the income distribution within the study population, an alternative approach could be to use the income quartiles of the entire Finnish population (all age groups) to demonstrate the income distribution within the study population relative to the national figures.

Response: While we appreciate the idea of using income quartiles from the entire Finnish population to illustrate the income distribution within our study population, we have chosen to maintain the proportions equal across the periods. This approach allows us to better identify the association between income and excess death within each period. If we were to use quartiles from the total population, the majority of individuals in our study population, being retired, would likely fall into the lowest income quartiles. By maintaining the proportions equal, we enhance the discriminatory power of our analysis and ensure a more accurate representation of the association between income and excess mortality within our study population. We hope this clarifies our approach, and we appreciate your thoughtful input.

4) It would be valuable to explore potential interactions between income and living arrangements, or between living arrangements and age, to gain a more comprehensive understanding of their effects on the study outcomes.

Response: Thank you for highlighting the potential value in exploring interactions between income and living arrangements, or between living arrangements and age, to gain a more comprehensive understanding of their effects on excess winter mortality. Given space constraints, we have only incorporated the recommendation regarding living arrangements and age as explained in response to comment number two. Regarding age, we would also like to draw attention to age-specific results provided in the Supplementary Material, in which differences by living arrangement were particularly modest among those in the oldest age category. This observation is also highlighted in the Results section (page 9) with the following statement “Additional age-category-specific results (Table S3) showed that while excess winter mortality increased with age, indications of effect modification by the analysed sociodemographic and health characteristics were mostly similar. Differences in the effect of winter were, however, more apparent among the younger age categories, whereas among 90-year-olds, no effect modification was observed.” Additionally, to explicitly acknowledge this aspect, we have included a statement in the manuscript's discussion section, emphasizing the importance of future studies to explore potential interactions between income and living arrangements. The added text (page 13) reads as: “Future studies should, however, assess possible interactions between income and living arrangements in diverse social settings to gain a more comprehensive understanding of their effects on excess winter mortality.”

5) In Table 3, there is an asterisk after "Rate per 1000 person years*.", if there are any additional information or context related to the rates, it is advisable to provide an explanation for this asterisk in the footnote.

Response: Thank you for noticing this. In Table 3 the asterisk was mistakenly used instead of the appropriate superscript 'a' to indicate that the winter and nonwinter rates have been age-adjusted. To provide a clearer presentation, we have removed the footnote and updated the column headings in Table 3, now clearly stating “Age adjusted” for the first two columns. We have also modified the column names of the last two columns, the title, and footnotes for improved clarity.

6) Table 3 presents the winter-nonwinter rate ratio in two models: one adjusted only for age (superscript a) and another adjusted for sex, age, income, living arrangement, region, and year (superscript b). A rationale for this distinction should be provided to justify why age alone was

considered in the first model while other sociodemographic characteristics were added in the second model.

Response: Thank you for bringing attention to this matter as upon reviewing your question, we recognized that there were errors in the column headings of Table 3. To clarify, the first two columns, representing winter and non-winter death rates, have been adjusted for age. This is now reflected in the column heading as explained in response to the previous question. On the other hand, the last two columns are adjusted for all sociodemographic variables, noted now with superscript a. We hope these corrections and modifications in the column headings and footnotes address your concerns.

7) Given the involvement of multiple analyses with different cohorts, it is recommended to include a Consort diagram or a table that delineates the various cohorts used in the relevant analyses. This will help the reader better understand the study's design and population selection.

Response: Thank you for your attention to the reporting of the study population selection. However, as our analyses are always based on the total population aged 60+ within a given period, we believe our population selection can be communicated within the narrative of the methods section without the need for a separate diagram. To enhance clarity, we have made the following modifications under the methods section (page 4): We restricted our analysis to individuals aged 60 years and over at the end of each year from 1971 to 2018.

8) Table S1 mentions the exclusion of alcohol-related conditions within mental health disorders, circulatory issues, and accidents and violence. To enhance transparency, it is advisable to detail this exclusion process in the methods section, and indicate whether any analyses of interactions within the pre-existing health conditions were conducted.

Response: We appreciate the opportunity to clarify the information regarding the categories shown in Table S1. Especially we want to clarify that alcohol-related diseases and causes were not excluded from our study but rather constitute a distinct category "Alcohol related causes" on their own, based on Statistics Finland's classification of causes of death. This specific categorization aligns with a broader initiative in Finland to better identify and understand alcohol-related causes of disease. For simplicity, we did not conduct interaction analyses within the pre-existing health conditions but assessed each condition separately.

9) The authors might consider assessing multi-morbidities of individuals rather than solely relying on dichotomous pre-existing health conditions. Alternatively, they should provide a justification for not considering multi-morbidities if that was the chosen approach.

Response: Thank you this interesting suggestion. We agree that exploring the impact of multiple pre-existing conditions simultaneously is an important and interesting aspect, as it may further increase vulnerability to excess winter mortality. Nevertheless, identifying the numerous specific sets of health conditions that, when present together, may be especially harmful falls beyond the scope of our current study. Treating multi-morbidities as a single category without acknowledging the potentially condition-specific effects might lead to unclear or even erroneous conclusions. We have thus chosen not to assess multi-morbidities in this study but acknowledge their importance as a significant avenue for further investigation.

10) In the analysis using multivariable Poisson regression to assess the risk of excess winter mortality, it is important to check for overdispersion in the model and report the findings accordingly.

Response: Thank you for highlighting this aspect. To account for the overdispersion the models were fitted with robust, clustered standard errors where each cluster was defined as an individual. We forgot to state this in the submitted version but have now added the following in the methods section (page 6): "To address potential issues of overdispersion in the Poisson regression models, robust, clustered standard errors were employed, with each cluster defined as an individual."

11) From the abstract and title, it has been stated that this is a prospective nationwide register study. However, in the methods section, it was mentioned, "This data was drawn from censuses and population registers..." and "All data was collected for routine administrative purposes..." If the data had already been collected before the initiation of the research, or were collected for other purposes but not specifically for the study, it would be more accurate to classify as a retrospective study. Therefore, please reconsider clarifying the study design type to ensure accuracy.

Response: We appreciate your attention to detail regarding the classification of our study. While we agree that the term "prospective" does not accurately characterize our study design, given the nature of our data source and collection methods, we also find that "retrospective" may not be an ideal descriptor. In survey settings, "retrospective" often implies asking individuals about past events, which doesn't align with our study's focus. To prevent potential misinterpretations, we have revised the title to: "Excess winter mortality in Finland 1971-2019 – a register-based study on long-term trends and effect modification by sociodemographic characteristics and pre-existing health conditions."

Reviewer: 4

Mrs. Cecilie Dahl, University of Oslo

Comments to the Author:

Dear Authors,

Thank you for providing this exciting and relevant paper for review. I would like to congratulate you. This is well-written paper on a topic that has not been much studied, and the results are important to elucidate the role of the cold climate in health and disease in the Nordic countries. It also takes advantage of already existing data for a very long time period, showing how important societal changes are in mitigating the effects from this harsh climate. I really enjoyed reading it, and my comments below, except for the last one about the interactions are only minor (to provide further clarity).

- First: An ethics statement is needed. I understand that consent is impossible, but did you obtain exception from consent for the registries? Did you need ethical consent from a committee for the linkages of the Census and the registries.

Response: The following ethics statement is included after the conclusion section: "Ethics approval: The use of the anonymized register data has been approved by Statistics Finland's Ethics Board. The register data are originally collected for administrative and statistical purposes. No informed consent is required for register-based studies in Finland."

- "Data" are plural. Therefore "data were" not "data was" (please change throughout the manuscript)

Response: These have been corrected.

- Table 3: are some category headings missing above yes and no at the end?

Response: We appreciate your observation regarding the unclear category headings. To enhance clarity, we have revised the headings for the last two pre-existing health condition categories. The updated names are now "Other than specified above" and "Any health condition". It's important to note that the change from the category "None" to "Any health condition" alters the meaning of the responses "Yes" and "No" within this category. With the new designation "Any health condition", "Yes" now signifies the presence of at least one health condition, and "No" indicates the absence of any specified health conditions. We hope this modification facilitates a more intuitive interpretation of the tables. These adjustments have been applied consistently across all tables and reflected in the corresponding text.

- How did you obtain the estimates for the effect modifications/interactions? Some more information is required under "statistics" in the method section. Usually, the interactions are tested (based on pre-defined hypotheses), then estimates are obtained by either stratifying on the variables of interest or predicted for each level of the variables (i.e., obtaining marginal effects) and lastly compared to obtain effect estimates. Note: the estimates that you obtain from the statistical interaction itself, or the model including the interaction term are not valid.

Response: Thank you for your inquiry regarding the estimation of effect modifications. We have chosen to present the interactions in multiplicative terms, following the approach outlined in Buis, 2010, and as previously applied in research on excess winter mortality, such as Wilkinson et al. 2004. This method aligns with our study objectives and offers a clear interpretation of how the effect of winter varies in relative terms between different categories of modifying factors.

In response to your suggestions, we have clarified the explanation of our approach in the methods section (page 6). The added text reads as follows: "These interaction effects, interpreted as relative risks, show how the effect of winter varies in multiplicative terms between different categories of modifying factors and the baseline reference group."

Furthermore, to enhance the readability of the tables, we have added footnotes to Tables 3, S3 and the new supplementary Table S4. The added footnote reads as follows: "Relative risks are the

interaction terms of the winter indicator and the effect modifier. These show the relative differences in the effect of winter on mortality between effect modifier categories.”

We hope these modifications provide a more comprehensive explanation of how our estimates were obtained and address your concerns.

References:

Buis, M. L. Stata Tip 87: Interpretation of Interactions in Nonlinear Models. The Stata Journal 2010;10(2):305-308. doi:10.1177/1536867X1001000211

Wilkinson P, Pattenden S, Armstrong B, et al. Vulnerability to winter mortality in elderly people in Britain: population based study. BMJ 2004;329:647. doi:10.1136/bmj.38167.589907.55

VERSION 2 – REVIEW

REVIEWER	Wong, Esther University of Bristol
REVIEW RETURNED	13-Dec-2023
GENERAL COMMENTS	Thank you for submitting the revised article; I appreciate your incorporation of some of my suggestions.
REVIEWER	Dahl, Cecilie University of Oslo, Department of Community Medicine and Global health
REVIEW RETURNED	19-Dec-2023
GENERAL COMMENTS	No further comments

VERSION 2 – AUTHOR RESPONSE